# SafeSoCPS: A Composite Safety Analysis Approach for System of Cyber-Physical Systems

**DOI:** 10.3390/s22124474

**Published:** 2022-06-13

**Authors:** Nazakat Ali, Manzoor Hussain, Jang-Eui Hong

**Affiliations:** Software Intelligence Engineering Laboratory, Department of Computer Science, Chungbuk National University, Cheongju 28644, Korea; nazakatali@chungbuk.ac.kr (N.A.); hussain@selab.cbnu.ac.kr (M.H.)

**Keywords:** cyber-physical systems, safety analysis, fault traceability

## Abstract

The System of Cyber-Physical Systems (SoCPS) comprises several independent Cyber-Physical Systems (CPSs) that interact with each other to achieve a common mission that the individual systems cannot achieve on their own. SoCPS are rapidly gaining attention in various domains, e.g., manufacturing, automotive, avionics, healthcare, transportation, and more. SoCPS are extremely large, complex, and safety-critical. As these systems are safety-critical in nature, it is necessary to provide an adequate safety analysis mechanism for these collaborative SoCPS so that the whole network of these CPSs work safely. This safety mechanism must include composite safety analysis for a network of collaborative CPS as a whole. However, existing safety analysis techniques are not built for analyzing safety for dynamically forming networks of CPS. This paper introduces a composite safety analysis approach called SafeSoCPS to analyze hazards for a network of SoCPS. In SafeSoCPS, we analyze potential hazards for the whole network of CPS and trace the faults among participating systems through a fault propagation graph. We developed a tool called SoCPSTracer to support the SafeSoCPS approach. Human Rescue Robot System—a collaborative system—is taken as a case study to validate our proposed approach. The result shows that the SafeSoCPS approach enables us to identify 18 percent more general faults and 63 percent more interaction-related faults in a network of a SoCPS.

## 1. Introduction

Cyber-Physical Systems (CPSs) are highly connected and massively networked systems of cyber (computation and communication) and physical (sensors and actuators) components that interact with each other in a feedback loop to achieve a common goal [1,2]. The System of Cyber-Physical Systems (SoCPS) is a complex, heterogeneous system comprising individual independent CPSs to achieve common goals that cannot be achieved by a single system [3,4]. The SoCPSs have significantly promoted the popularity of many emerging intelligent systems in our daily lives, for instance, smart agriculture, smart grids, robotic systems, intelligent transportation, and avionic collision avoidance systems. These SoCPS are often part of wider collaborative networks consisting of many other CPSs and form networks capable of providing functionalities that individual systems cannot provide. A platooning system, for instance, allows vehicles to reduce intervehicle distance and hence save fuel as a common speed of all vehicles in the platoon is negotiated by all adaptive cruise control units involved. These types of system networks are highly dynamic as the individual systems can join and leave such networks at runtime.

SoCPS offer unprecedented opportunities to monitor and control the physical world through computation and control functionalities. However, these complex systems pose numerous safety-related challenges because any failure modes within a network of SoCPS may have profound effects on the whole system. Therefore, SoCPS need an extensive rule of design and adherence to the safety properties for all possible interactions so that the opportunities offered by these systems are fully welcomed while ensuring safety. During the operation, if one participant system fails, it impacts the final goal of the SoCPS. Therefore, composite hazard analysis is required where the network of a SoCPS will be analyzed to know the system’s potential failures and their impact on other collaborative systems during design time. As mentioned, the SoCPS are networks of CPSs; therefore, hazard analysis for a single system cannot guarantee SoCPS’s safety due to collaborative behavior. Another major challenge for a network of collaborative CPSs is the validation of behavior emerging due to the collaboration of networked CPSs. This potential emerging behavior can either be a desired property of a system that can enable some kind of functionality in a SoCPS or an undesired event that can potentially lead to a dangerous state. Daun et al. [5] contributed to the review of the emergent behavior in the networks of CPSs by proposing the automated generation of instance models that allow the assessment of different network configurations of SoCPS. The proposed approach depends on the automated generation of diagrams to validate different configurations of a SoCPS considering multiple instances of participating CPSs.

However, automated safety analysis techniques can aid in the validation of expected behaviors of a system [6,7]. To ensure safety for a network of a SoCPS, the individual participant system’s behavior for each function should be analyzed together with the collective behavior of other participant systems. Analyzing safety for each participant system in SoCPS cannot guarantee the safety of the whole SoCPS. It must be ensured that the network CPSs behave safely, which means that it is necessary to identify the safety faults that arise from the interplay between different CPSs in SoCPS. Hence, it is insufficient to ensure the correct behavior of each system, as the behavior stems from the interaction of various systems that cannot be attributed to an individual system and, therefore, cannot be specified for an individual participant system. SoCPS is an integrated set of systems that uses each system in a collaborative fashion to achieve a common mission that the individual systems in the network of SoCPS cannot achieve. Moreover, the SoCPS employ interdependencies that further complicate the system’s operations.

Traditional safety analysis techniques cannot cope with the complexity of SoCPS because each system in SoCPS is considered to be an independent system, and its safety analysis is also conducted independently. Therefore, new safety analysis techniques are required to analyze hazards for collaborative SoCPS. Furthermore, these new safety analysis techniques must handle a large network of SoCPS and produce meaningful results while remaining economically practical. Therefore, an automated composite hazard analysis would support the safety of the dynamically forming network of CPS. Our proposed automated composite safety analysis technique relies on individual CPSs’ available documentation (hazard analysis artifacts) and predefined constraints for collaboration among CPSs. To this end, we developed a tool called SoCPSTracer that takes hazard analysis artifacts for participant collaborative CPS as input and generates a Fault Propagation Graph (FPG). It is a directed graph that enables safety engineers to determine the flow of faults in the network of SoCPS. The FPG also gives information about inter-system and intra-system fault propagation and their impact on other systems in the network of SoCPS. The SoCPSTracer tool mainly contributes to the traceability strategy that defines a traceability information model, processes, and tooling in traceability fundamentals [8,9]. In summary, we make the following contributions to this article:First, we define three relationships (i.e., influence, countermeasure, and overlap relationship) among the hazard analysis artifacts of participating systems in SoCPS. The relationship among the faults, their causal factors, outcomes, and countermeasures provide information about the propagation of a fault within a single system and/or in the network of SoCPS. We assume that all participating CPSs in SoCPS are analyzed using FMEA, ETA, and FTA. We also assume that hazard artifacts for all participating systems in a network of SoCPS are available. However, we can also produce hazard artifacts for participating systems using our tool and make them available.Second, to support safety engineers in analyzing and investigating the potential faults, we propose a diagrammatic representation of identified faults and their manifestations in the SoCPS network. The automatically generated diagrammatic representation shows fault propagation in the network of SoCPS, which we call FPG. Thus, the FPG gives information about the propagation of a fault in the network of SoCPS. Using this FPG, we can trace faults back to their source and apply preventive mechanisms to mitigate the potential faults.Third, a tool called SoCPSTracer is developed to support our SafeSoCPS approach. This proposed tool enables safety engineers to analyze safety for a network of CPSs and produce a FPG that determines the propagation of faults among/between systems and their impact on other components/systems.

Thus, the remainder of this paper is organized as follows. Section 2 discusses the background and related work. In addition, Section 3 illustrates the proposed approach. Section 4 presents a tool to support our proposed approach, and Section 5 illustrates a case study to validate our proposed approach. Finally, the limitations of this study of are presented in Section 6, followed by a summarizing conclusion in Section 7.

## 2. Background and Related Work

### 2.1. System of Cyber-Physical Systems

In this article, we focus on SoCPS, and we provide a distinction between CPSs, System-of-Systems (SoS), and SoCPSs. CPSs are the integrations of computation, networking, and physical components of the system [10]. SoCPS is a set of individual CPSs which can offer new functionalities due to their collaborative capabilities [3]. By definition, SoCPSs are similar to SoS. The concept of SoCPS arises after the emergence of CPSs [11], whereas the concept of SoS is very old [12]. Standard practice for system safety, i.e., MIL-STD-882E defines the term system-of-systems as “a set or arrangement of interdependent systems that are related or connected to provide a given capability” [13]. According to ISO 21,841 standard, SoS are a “set of systems or system elements that interact to provide a unique capability that none of the constituent systems can accomplish on its own” [14]. In our context, the constituent systems are independent CPSs that form a network of SoCPSs. Maier [12] has categorized SoS into three categories: (1) Directed SoS, a master system that coordinates the slave systems. (2) Virtual SoS, a system with no agreed purpose or the central management, and (3) collaborative SoS, where participating systems may form a coalition to accomplish a common goal. Our concept of SoCPSs is similar to the idea of collaborative SoS where a network of CPSs is formed when two or more CPSs collaborate to accomplish a common goal that an individual system could not achieve. To that end, the participant CPSs exchange various information, e.g., vehicles in the platooning system share their speed and intervehicle distance to achieve their collective goal. The SoCPSs is a group of largely pre-existing CPSs that may be independently owned and managed but collectively are relied upon to do a task that is not possible without collaboration [11]. For instance, the distributed energy resource, a small-scale system that consumes, stores, and/or produces energy in the smart grid domain, can collaborate and make a virtual power plant to provide energy-related services to the market [15].

### 2.2. Safety Analysis for System of Cyber-Physical Systems

Safety is an essential need of CPSs that can address accidental faults and failures in both individual systems and a CPSs network. Therefore, from the perspective of safety engineering, all the potential interactions the individual participating systems might encounter should be considered. These interactions among/between systems pose a serious safety threat for the collaborative systems. For example, Harvey and Stanton [16] have conducted a literature review in the domain of safety for SoS and identified ten key challenges, including complexity, emergent behavior, unpredictability, and safety.

Baumgart et al. [17] addressed the concept phase for developing a SoS, where authors evaluated different technical concepts for a specific product feature. The authors utilized automated guided vehicles called HX as a case study to validate their approach. The HX was used to transport raw materials from a movable primary crusher to a stationary secondary crusher in an open-surface mine. Human operators were responsible for loading the HX. The authors identified potential hazardous scenarios and proposed an approach called SafeSoS to analyze the potential hazards for identified hazardous scenarios. The authors documented the behavior of SoS using SysML sequence diagram. The information from the sequence diagram was transferred automatically to feed to Failure Mode and Effect Analysis (FMEA) and Hazard and Operability Analysis (HAZOP) hazard analysis techniques, respectively. The authors identified interface hazards, resource hazards, and interoperability hazards during the hazard analysis phase for SoS. Under SafeSoS research, the authors [18] proposed a hierarchical process to document the interactions of SoS.

Saberi et al. [19] took connected vehicles as an application domain of SoS and proposed a tailored safety lifecycle by adding required activities for SoS safety analysis. The authors defined functions, system definition granularity, and use cases at the vehicle and connected vehicle levels. In parallel to the hazard analysis for a single system, hazard analysis for SoS was performed to identify the hazards that may emerge from the collective behavior of SoS. The authors also developed a safety concept for SoS to discover any safety measure required to mitigate hazards at the ‘connected vehicle level’ along with the definition of the functional safety concept for a single vehicle. At this stage, Fault Tree Analysis (FTA) supported the hazard analysis activity. When the safety verification at the vehicle level was conducted, an additional level of safety verification was also carried out to test the correct execution of the safety mechanisms at the connected vehicles’ level. The authors argue that these two levels of safety verification can ensure safety for SoS. They also presented a process for ensuring safety for a SoS. However, this study did not consider collective hazard analysis for a SoS, and it also does not provide the automated tool support to foster the safety analysis process.

Kochanthara et al. [20] have investigated to check whether the architecture for a single vehicle (a participant system in SoS) meets the functional safety requirements for cooperative driving (an SoS). The authors proposed a method to assess functional safety for cooperative vehicles by combining software architecture domains and safety engineering methods. They first derived functional safety requirements for cooperative vehicles and then checked whether the designed software architecture for cooperative driving meets the derived functional safety requirements. The authors’ main focus was on the development phase in ISO 26,262 and the validation of the resulting requirements in the software architecture in the final product. However, this study did not consider fault identification and traceability in SoS.

Causevic et al. [21] have proposed a framework in order to develop a safe and secure adaptive collaborative system with runtime guarantees. In the proposed framework, the authors focused on requirements engineering and safety assurance techniques in order to capture safety and security properties for collaborative systems. The authors also provided safety assurance cases which provide a guarantee that the system is sufficiently safe. Furthermore, the authors also proposed architecture and behavioral models to analyze safety requirements at runtime.

Table 1 shows the summary of related work and its comparison with our SafeSoCPS approach.

## 3. SafeSoCPS Approach

Generally, safety analysis is often considered a combination of manual and automated techniques [22,23]. This section explains our approach called SafeSoCPS where we follow a general four-step approach for safety analysis in CPSs. First, the safety requirements for the system under development are defined. Second, the potential safety hazards for participant systems and for SoCPS are identified using a composite hazard analysis. Though various safety analysis techniques exist, we assume that safety engineers conduct Failure Mode and Effect Analysis (FMEA), Event Tree Analysis (ETA), and Fault Tree Analysis (FTA) to identify faults, consequences, and sources of identified faults. Third, based on the safety analysis results obtained in the second step, the requirements are modified or added new requirements (first step) and repeated in the second step. Fourth, the newly obtained knowledge about the occurrence and the manifestation of potential faults is considered to discover new faults, and the system is lastly certified for safety. The SafeSoCPS approach contributes mainly to the second step of the general safety process that we follow.

In the following subsections, we explain composite safety analysis for SoCPS (SafeSoCPS).

### 3.1. Content Relationship among Hazard Analysis Artifacts

Content relationships among hazard analysis artifacts for multiple CPS are defined to support the safety analysis for the network of a SoCPS. Initially, we introduced content relationship between hazard analysis artifacts for FTA, ETA and FMEA in [24]. However, in this article, we present an improved version of those defined relationships. The description of these relationships is as follows:

Influence Relationship: A relationship in which a fault of one participating CPS affects another participating system (s) in the SoCPS network.

Overlap Relationship: If a failure mode or a causal factor in FMEA and an initiating event in ETA lead to the same consequences. Then, the consequences (system effect in FMEA and outcomes in ETA and vice versa) will have an overlapping relationship. Simply, if two faults in the network of SoCPS result in the same consequences, then their consequences will have an overlapping relationship.

Countermeasure Relationship: A countermeasure relationship exists when the safety guard for a particular fault in one participating CPS is used to counter a fault (s) in another participating CPS in the network of SoCPS.

Figure 1 shows an example relationship among hazard analysis artifacts. For instance, a failure mode in FMEA, i.e., “Detection Failure” has an “Influence Relationship” with “Robot Collision”, an outcome in ETA, and “Robot Collision” event in FTA. Similarly, the “Robot Collision”, a system effect in FMEA has an “Overlap Relationship” with “Robot Collision” which is an outcome of fault “Obstacle Detection Failure”. Meaning that both faults “Detection Failure” in FMEA of searching robot (a participating CPS in HRRS) and “Obstacle Detection Failure” in ETA of obstacle robot led to the same consequence “Robot Collision”. We see that the safety guard for the “Obstacle Detection Failure” fault in ETA is not known. However, we observe that the “Detection Failure” fault which overlaps “Obstacle Detection Failure” has a safety guard, i.e., “Increase Sensor Capability”, to avoid robot collision. Therefore, an “Increase Sensor Capability” safety guard can be used as a countermeasure to mitigate the “Obstacle Detection Failure” fault in ETA.

### 3.2. Composite Safety Analysis Model

In order to realize the above defined relationships, we formalized the hazard analysis techniques and proposed a Composite Safety Analysis Model (CSAM). The definition of CSAM is as follows:

Definition 1 (Safety Analysis for SoCPS): Safety analysis for SoCPS is defined as a tuple SA = <ID, HAT, S, L>. Where ID is a unique identification for safety analysis, HAT is a hazard analysis technique applied to analyze the system S ϵ SoCPS such that {∀ TC ∈ { FMEA, ETA, FTA}∈ HAT and L is the relationship among the components of hazard analysis artifacts.

Definition 2 (FMEA)*:* FMEA model is defined as a tuple of FMEA = <ID, I, FM, SE, CF, RA, L>. Where ID is a unique identification of FMEA set, I is a set of item/function lists, {i_1_, i_2_, …}, FM is a set of failure modes, {fm_1_, fm_2_, …}, SE is the set of effects/hazards, {se_1_, se_2_, …se_n_} in FMEA, CF is the set of causal factors, {cf_1_, cf_2_, …, cf_n_}, RA is the set of recommended actions/safety guards, {g_1_, g_2_, …, g_n_} provided for a particular fault in FMEA, and L is the established relationship link of a component C = (I, FM, SE, CF, RA) with other components in FTA and ETA such that C ∈ FMEA → C ∈ FTA V C ∈ ETA. Meaning that the components which belong FMEA may have a relationship with the components of FTA or ETA.

Definition 3 (FTA): FTA model is defined as a tuple of FTA = <ID, G, E, TE, L>, where ID is the unique identification of the FTA set, G is the set of gates in a fault tree, E is the set of event modes E {e_1_, e_2_, … e_n_} in the fault tree, TE is the set of top event TE {te_1_, te_2_, …te_n_} in a fault tree, and L is the established relationship link of a component C = (E, TE) with other components in ETA and FMEA such that C ∈ FTA → C ∈ ETA V FMEA.

Definition 4 (ETA): ETA model is defined as a tuple of ETA = <ID, I_E_, P_E_, O, L>, where ID is a unique identification of ETA set, I_E_ is set the initial events I_E_{ i_e1_, i_e2,_ …, i_en_}, P_E_ is the set of pivotal events P_E_ {pe_1,_ pe_2,_ …, pe_n_} in ETA, O is the set of outcomes {o_1_, o_2_, …, o_n_} of an initiating event I_E_ of an ETA, and the L is the established relationship link of a component C = (I_E_, P_E_, O) with other components in FTA, and FMEA such that C ∈ ETA → C ∈ FTA V FMEA.

### 3.3. Diagrammatric Representation

The diagrammatic representation in SoCPS is critical because the information about fault propagation in the network of CPS can help safety engineers to mitigate a particular fault. This information may also help safety engineers to identify potential faults present in the participating CPSs and their propagation route. After identifying faults and their propagation routes, safety engineers can generate behavioral models for different CPS network configurations and provide them as input to the system verification techniques. The behavioral models of participating CPSs in SoCPS network configurations can be checked for the unwanted behaviors identified during a composite hazard analysis. Identifying the unwanted behavior during a specific configuration may help to correct the cause of the undesirable behavior in the SoCPS. However, in SoCPS, the unwanted behavior may occur due to the interplay of the systems, which cannot be corrected easily because the interaction among systems is dynamic in nature. The behavioral models for an overall network of SoCPS should be investigated to fix unwanted behaviors. Therefore, to support safety engineers, there is a need to trace unwanted behavior within a participating system and in the network of a SoCPS. Figure 2 shows the traceability of a fault within a participating system to other participating CPSs in the network of SoCPS.

Highlighting the propagation of a fault in the original specification (Figure 2) can help to investigate the manifestation of faults in the collaboration of different systems in the network of the SoCPS. However, since the SoCPS is a complex network of individual systems; manual analysis of such diagrams is time-consuming and economically not feasible. Therefore, we developed a tool called SoCPSTracer which produces the fault propagation graph to know the faults propagation and their impacts in the network of SoCPS.

The proposed FPG is a directed graph G = (N, E), where N is a node in FPG such that N ∈ C in FMEA V ETA V FTA and E represents edges where each e of E is specified by n ordered pair of nodes n_1_, n_2_ ∊ N. The edge between e = (n_1_, n_2_) shows the edge (relationship) e between node n_1_, and n_2_ which can also be written as n_1_→n_2._ Further explanation of FPG is mentioned in Section 4.

## 4. SoCPSTracer

The SoCPSTracer is an exclusively developed tool to support the SafeSoCPS approach. The SoCPSTracer comprises three major components, i.e., safety analysis manager, traceability analyzer, and traceability presenter, as shown in Figure 3. SoCPSTracer is implemented in Java and JavaFx was used to develop user interface. The system was equipped with core i7 processors and 32 GB RAM for the experimental setup. Additionally, NVIDIA GeForce RTX2060 GPU was added to the system to faster the computational process, and better visualization of data on FPG. Data visualization on FPG is supported by an opensource java library called smartgraph (https://github.com/brunomnsilva/JavaFXSmartGraph, accessed on 20 April 2022) that supports directed and undirected graph generation. We customized the smartgraph library to visualize the data on FPG according to our requirements.

The major components of SoCPSTracer are described as follows:

Safety Analysis Manager: This component comprises four sub-components, i.e., FTA editor, ETA editor, FMEA editor, and importer, as shown in Figure 4 (left, 1). The participant systems in the network of SoCPS can be analyzed using the respective hazard analysis techniques and produce hazard analysis artifacts for participating systems. The produced artifacts are then saved into the hazard analysis artifacts repository. The importer can be used to import already existing hazard analysis artifacts for participating systems in the network of SoCPS.

Traceability Analyzer: This component comprises a relation detector and traceability repository, as shown in Figure 4 (top, 2). A relation detector is a component that identifies and connects the trace links (relationship) among hazard artifacts. Algorithm 1 is used to detect relationships among the hazard analysis artifacts. The traceability repository is used to store trace models.

Traceability Presenter: This component of SoCPSTracer is consists of a traceability viewer sub-component and a impact analysis sub-component. The traceability viewer displays visual trace information among hazard analysis artifacts which we call FPG in the SoCPSTracer, as shown in Figure 4 (top, 3). FPG is the manifestation of the relationship among hazard analysis artifacts. The impact analysis in FPG helps to determine the impact of a fault on other participating systems.

The FPG is a digraph of vertices and edges. The edges connect the vertices or nodes in FPG. The nodes represent faults or safety guards, whereas edges represent the relationship between two or mode nodes. An edge is placed between a pair of nodes if they are related to each other in a certain way.

Formally, the FPG is expressed as, Let G = ( N, ${\mathrm{ID}}_{n}$, E) where N is a finite set of FPG nodes, i.e.,$\;N\;\left(n_{1},n_{2,...,}n_{m}\right)$. Each node n in FPG carry some information to describe the node, i.e., n = (${\;n}_{\mathrm{description}},S_{\mathrm{name},}{\mathrm{HAT}}_{\mathrm{name}}\;$), where $n_{\mathrm{description}}$ is the description of a fault or a safety guard, $S_{\mathrm{name}}$ is the name of the system from where a fault or safety guard belongs to, and ${\mathrm{HAT}}_{\mathrm{name}}$ is the name of the hazard analysis technique that is used to analyze that particular node. The E in G represents edges where each e ∊ E is specified by n ordered pair of nodes n_1_, n_2_ ∊ N. The edge between e = (n_1_, n_2_) shows the edge *e* between vertex n_1_ and n_2,_ which can also be written as n_1_→n_2_. In order to check the fault propagation of a specific fault in FPG, we use a subgraph that considers only influence relationships in FPG and generates the propagation graph for a specific fault. In order to know the fault propagation for a particular node in FPG, it is necessary to find the in-degree and out-degree of that specific node.

Let ${\mathrm{FPG}}_{\mathrm{out}}$ be a graph for the propagation of a specific node n ∈ N. The neighbors of n are the set of nodes adjacent to n through an influence relationship. Therefore, the out-degree of a fault in FPG is used to draw ${\mathrm{FPG}}_{\mathrm{out}}$.

${\mathrm{FPG}}_{\mathrm{out}}$ (n) = {n ∈ N: ∃e ($R_{\mathrm{influence}})$ ∈ E (e = {n_1_, n_2_} or n_1_ = n_2_ and e = {n_2_})} where ${\mathrm{FPG}}_{\mathrm{out}}$(n) is the set of nodes n ∊ N such that there exists an edge *e* so that the influence relationship always holds in E so that e = {n_1_, n_2_} holds.

Similarly, the in-degree of a specific fault in FPG is used to recover the traceability of a specific fault which is called ${\mathrm{FPG}}_{\mathrm{in}}$. ${\mathrm{FPG}}_{\mathrm{in}}$ is for a particular fault, and *n* tells what kind of other faults may lead to that particular fault n. The set of incoming edges of a vertex, e.g., n_1_ ∊ N are all those edges whose arrows point into n_2_ ∊ N.

${\mathrm{FPG}}_{\mathrm{in}}$ (n) = {n ∈ N: ∃e ($R_{\mathrm{influence}})$ ∈ E (e = (n_1_, n_2_) | n_2_, n_1_))}.

Each n ∈ N has a unique ID given by ${\mathrm{ID}}_{n}\;\left(n\right)$. The relation *e* gives the communication topology between nodes n_1_ and n_2._ An edge e = ($n_{1},{\;n}_{2}$) indicates a directed relationship between $n_{1}$ and $n_{2}$.

In FPG, every node *n* ∊ *N* contains information that helps safety engineers understand the relationship of a specific fault that belongs to a particular system and is analyzed by a particular hazard analysis technique. Therefore, node *N* = (*system, elementOf, description*) has three kinds of information: the *system* which tells from which system the fault or safety guard belongs to, *elementOf* uniquely determines the faults or safety guards belonging to which hazard analysis technique and *description* is the definition of faults or safety guards. This information helps to trace a fault among a network of SoCPS.

Let X (faults from FTA, FMEA, and ETA) are the sets of faults we are interested in discovering their relationships, i.e., countermeasure relationship (R1), influence relationship (R2), and overlap relationship (R3). Let Z is the disjoint union set of X. Therefore, Z {x: x ∈ X}. Let A {a_1_, a_2,_ ..., a_m_} is the set of hazard analysis artifacts, i.e., failure modes, causal factors, system effects and recommended actions obtained from FMEA, top events, intermediate events, and basic events obtained from FTA, and pivotal events, initiating events and outcomes obtained from ETA. Let *A* is the disjoint union set of A. Therefore, A{*a*: *a* ∈ A } and A ⊂ Z.

The influence relationship (R2) can be established in three perspectives. Therefore, in Algorithm 1, each influence relationship is reflected separately. In the first case, n (failure mode) and m (causal factor) may have an influence relationship with a hazard artifact *a_m_* such that n and m lead to x (system effect), which belongs to FMEA if and only if the *x* is determined to be similar with a_m_ (event) in FTA or a_m_ (outcome) in ETA. The similarity between two contents of the hazard analysis techniques is calculated using Jaccard similarity index which is a commonly used algorithm to compare two strings [25]. Equation (1) shows the Jaccard similarity index where J(A,B) is the Jaccard similarity between string A and string B.
(1)$$J\left(A,B\right)=\frac{\;\left|\;A\cap B\right|}{\;\left|\;A\cup B\right|}$$

In the second case, the y (initiating event) may have an influence relationship with a hazard artifact a_m_ such that y leads to x, which belongs to ETA if and only if x is determined to be similar with a a_m_ (event) in FTA or a_m_ (system effect) in FMEA. In the third case, the w (child events) may influence a_m_ such that w led to x, which belongs to FTA if and only if x is determined to be similar to a_m_ (system effect) in FMEA or a_m_ (outcomes) in ETA.

Two kinds of possibilities may exist for the countermeasure relationship (R1), as mentioned in Algorithm 1. In the first case, the c (recommended action in FMEA) counters e (event in FTA) such that c is defined as recommended action for x (failure mode) in FMEA if and only if x is determined to be similar to e in FTA. In the second case, the c counter h (initiating event) such that c is defined as recommended action x (failure mode) in FMEA if and only if x is determined to be similar to h (initiating event in ETA).

Algorithm 1 also shows that there may exist an overlapping relationship (R3) between x (system effect) in FMEA such that u (failure mode) lead to x in FMEA, if and only if x is determined to be similar to a_m_ (outcome) in ETA and vice versa.
**Algorithm 1:** FPG generation for fault traceability
**Input**: R1, R, R3
**Output:** FPG1Z {a : a ∈ A }2R1(c, e) ∨(c , h)←∅ of Countermeasure Relationship3$R2\left(\left(n\;\wedge \;m\right),{\;a}_{m}\right)\;\vee \;\left(y\;,{\;a}_{m}\right)\;\vee \;\left(w\;,{\;a}_{m}\right)\leftarrow \emptyset \;\mathrm{of}\;\mathrm{Influence}\;\mathrm{Relationship}$4$R3\left(x,{\;a}_{m}\right)\leftarrow \emptyset \;\mathrm{of}\;\mathrm{Overlap}\;\mathrm{Relationship}$5 Foreach a ∈A6
Foreach x ∈Z do7

$R1\left(c\;,\;e\right)\leftarrow \left\{\exists x:\;c\mapsto \;x\;\in \mathrm{FMEA},\;\mathrm{has}\;≅\;i.e.\frac{\;\left|\;x\cap \left(e⇢a_{m}\;\in A\in \;\mathrm{FTA}\right)\right|}{\;\left|\;x\cup \left(e⇢a_{m}\;\in A\in \;\mathrm{FTA}\right)\right|}\;\mathrm{to}\;\left(a_{m}\in A\in \mathrm{FTA}\right)\right\}$8

$R1\left(c\;,\;h\right)\leftarrow \left\{\exists x:\;c\mapsto x\in \mathrm{FMEA},\;\mathrm{has}≅i.e.\frac{\;\left|\;x\cap \left(h⇢a_{m}\in A\in \;\mathrm{ETA}\right)\right|}{\;\left|\;x\cup \left(h⇢a_{m}\in A\in \mathrm{ETA}\right)\right|}\;\mathrm{to}\;\left(a_{m}\in A\in \mathrm{ETA}\right)\right\}$9

$R2\;\left(\left(n\;\wedge \;m\right),{\;a}_{m}\right)\;\;\;\;\;\;\;\leftarrow \left\{\;\exists x:\;n\;,\;m⇢x\in \mathrm{FMEA}\;\mathrm{has}\;\;\;\;\;\;\;≅i.e.\;\frac{\;\left|\;x\cap \left(a_{m}\in A\in {\mathrm{FTA}\;V\;a}_{m}\in A\in \mathrm{ETA}\;\right)\right|}{\left|\;x\cup \left(a_{m}\in A\in {\mathrm{FTA}\;V\;a}_{m}\in A\in \mathrm{ETA}\;\right)\right|}\;\geq \mathrm{threshold}\;\mathrm{to}\;\left({\;a}_{m}\in A\;\;\;\;\;\;\;\in {\mathrm{FTA}\;V\;a}_{m}\in A\in \mathrm{ETA}\right)\right\}$10

$R2\;\left(y,{\;a}_{m}\right)\leftarrow \left\{\exists x:\;y\;⇢x\in \mathrm{ETA}≅i.e.\;\frac{\;\left|\;x\cap \left(a_{m}\in A\in {\mathrm{FTA}\;V\;a}_{m}\in A\in \mathrm{FMEA}\right)\right|}{\left|\;x\cup \left(a_{m}\in A\in {\mathrm{FTA}\;V\;a}_{m}\in A\in \mathrm{FMEA}\right)\right|}\;\;\;\;\;\;\;\;\geq \mathrm{threshold}\;\mathrm{to}\;\left(a_{m}\in A\in {\mathrm{FTA}\;V\;a}_{m}\in A\in \mathrm{FMEA}\right)\right\}$11

$R2\;\left(w,{\;a}_{m}\right)\leftarrow \left\{\exists x:\;w\;⇢x\in \mathrm{FTA}≅i.e.\;\frac{\;\left|\;x\cap \left(a_{m}\in A\in {\mathrm{FMEA}\;V\;a}_{m}\in A\in \mathrm{ETA}\right)\right|}{\left|\;x\cup \left(a_{m}\in A\in {\mathrm{FMEA}\;V\;a}_{m}\in A\in \mathrm{ETA}\right)\right|}\;\;\;\;\;\;\;\;\geq \mathrm{threshold}\;\mathrm{to}\;\left(a_{m}\in A\in {\mathrm{FMEA}\;V\;a}_{m}\in A\in \mathrm{ETA}\right)\right\}$12

$R2\;\left(w,{\;a}_{m}\right)\leftarrow \left\{x\;\;\;\;\;\;\;⟺a_{\;m}\;\left(\mathrm{if}\;\exists x:v⇢x\in \;\mathrm{FMEA}\;\mathrm{has}\;≅i.e.\;\frac{\;\left|\;x\cap \left(v⇢a_{m}\in A\in \mathrm{ETA}\right)\right|}{\;\left|\;x\cup \left(v⇢a_{m}\in A\in \mathrm{ETA}\right)\right|}\;\;\;\;\;\;\;\geq \;\mathrm{threshold}\;\mathrm{to}\;\left(a_{m}\in A\in \mathrm{ETA}\right)\right)V\left(\mathrm{if}\;\exists x:u⇢x\in \mathrm{ETA}\;\mathrm{has}\;\;\;\;\;\;\;\;≅i.e.\;\frac{\;\left|\;x\cap \left(u⇢a_{m}\in A\in \mathrm{FMEA}\right)\right|}{\;\left|\;x\cup \left(u⇢a_{m}\in A\in \mathrm{FMEA}\right)\right|}\;\;\;\;\;\;\;\geq \;\mathrm{threshold}\;\mathrm{to}\;\left(a_{m}\in A\in \mathrm{FMEA}\right)\right)\right\}$13
end14end15FPG ← R1(c,e) V (c,h) + R2 ((n Λ m), $a_{m}$) V (y, $a_{m}$) V (w, $a_{m}$)) + R3 ($x,{\;a}_{m}$)

## 5. Evaluation

The overreaching goal of SafeSoCPS is to ensure safety for collaborative CPSs, i.e., SoCPS using composite safety analysis while providing safety guards for identified faults where necessary. To evaluate our proposed approach, we investigate the applicability and usefulness of the SoCPSTracer for SoCPS. Therefore, we report the application of composite safety (SafeSoCPS) to the Human Rescue Robot System (HRRS), a SoCPS, and also conducted general safety analysis for individual participating systems to see the superiority of our proposed approach. Thus, we look at the following research questions.

**RQ1**: Can the SafeSoCPS approach identify more faults for collaborative CPSs?**RQ2**: Can the SoCPSTracer tool provide automated fault traceability among/between system components and participating collaborative systems?

### 5.1. Case Study

In hazardous situations such as fire or earthquakes, the rescue teams must deal with extremely hazardous and dangerous tasks. For instance, on 24 June 2021, portions of the surfside’s twelve-story Champlain Towers South condominium collapsed, killing 98 people, and injuring many. As a rapid response, Miami Dade Fire Rescue (MDRF) primarily carried out tactical drone operations for direct lifesaving and mitigation activities [26]. The rescue robots are supposed to perform safe and complex operations in order to rescue victims from the disaster. In our case study we consider HRRS (a SoCPS) which consists of a control station (CS), a searching robot (SR), an obstacle-removing robot (OR), and a lifesaving robot (LSR), as shown in Figure 5.

The robots, along with the CS, collaborate to perform rescue operations. The CS triggers rescue operations by sending disaster location area coordinates to SR. It also monitors and controls overall rescue operations. The SR obtains the coordinates of the place of interest and navigates to the disastrous location and starts searching for the victims by moving around the provided coordinates. Next, the SR sends the location of the victims to the OR. The OR obtains the location of victims and navigates to the location of victims and removes the obstacles around the victims and sends a message of clearance to the LSR. Finally, the LSR robot navigates to the location of the victims and moves them to a safe place.

The HRRS is a collaborative system where a single failure in the system may lead to the mission’s failure. Each system in the HRRS closely collaborates with each other; therefore, a fault in one system can propagate into the whole HRRS and may lead to serious hazardous scenarios. Thus, to mitigate the fault in the entire HRRS, we need to trace the faults back to their source. For instance, if the CS sends the wrong coordinates to robots in the field, it may result in the wrong navigation of SR, OR, and LSR; and as a result, it leads to mission failure, as shown in FPG (Figure 6, left bottom).

### 5.2. Results

In order to check the applicability and usefulness of our proposed approach, we first analyzed each system, i.e., CS, SR, OR, and LSR, using FTA, FMEA, and ETA and saved the artifacts in the hazard analysis artifacts repository. After receiving the hazard artifacts, we used them as input to the SoCPSTracer to generate FPG. An excerpt of FPG for HRRS is shown in Figure 6, where the red, yellow, and pink edges between nodes indicate countermeasure, overlap, and influence relationships, respectively. The FPG in Figure 6 shows inter-system fault propagation and intra-system fault propagation. For instance, the fault *Incorrect Coordinates.(CS.FMEA_3)* in CS leads to *Wrong Navigation.(SR.FTA_2)* in SR, and it further leads to *Mission*
*Failure.(HRRS,FMEA_4)* by going through Wrong *Navigation.(OR.FMEA_1)* in OR and Wrong Navigation.(LSR.FMEA_1) in LSR. This shows inter-system fault propagation because a fault *Incorrect Coordinates.(CS.FMEA_3)* that belongs to CS may lead to *Mission Failure.(HRRS,FMEA_4)* by propagating into SR, OR and LSR. The FPG can be a complex graph with thousands of edges. In such a case, manual fault propagation analysis becomes impossible (In our example, we deliberately kept FPG small and simple for better visibility and understandability); therefore, the subgraphs (${\mathrm{FPG}}_{\mathrm{out}},{\;\mathrm{FPG}}_{\mathrm{in}}$) of an FPG help in investigating the fault traceability for a specific fault. For example, if we want to check the fault propagation route for a particular fault, i.e., *Wrong Navigation.(SR.FTA_2)*, we double click on it, and it gives the fault route for that specific fault, as shown in Figure 7a. This subgraph shows the impact of one fault on another system (s).

During impact analysis, if we want to see what kinds of faults led to a particular fault, we can double click on that particular fault for back traceability. For instance, if we are going to see what kinds of faults led to *Mission Failure.(HRRS,FMEA_4)* in Figure 7a, we double click on it and receieve another subgraph ($FPG_{in}$) as shown in Figure 7b. We see that *Wrong Navigation.(SR.FTA_2)* and *Wrong Navigation.(LSR.FTA_1)* led to the *Mission Failure.(HRRS,FMEA_4)* fault. These kinds of graphs trace the faults back to their route causes. After tracing the faults, safety engineers can find the fault’s route cause, prioritize the criticality of faults, and design/use safety countermeasures to mitigate those faults. Therefore, in response to **RQ2**, we can say that SoCPSTracer tool supports fault traceability for collaborative CPSs as reflected in Figure 6 and Figure 7.

### 5.3. Discussion of Results

The influence relationship enables safety engineers to trace faults in the generated FPG, whereas the overlap relationship provides information on what kinds of faults lead to the same consequences. For instance, *Mission Failure.(HRRS,FTA_3)* and *Mission Failure.(HRRS,FMEA_4)* are the same consequences of *Robot Failure.(SR.FMEA_0)*, *Robot Failure.(LSR.FMEA_2)*, *Wrong Navigation.(SR.FMEA_0)*, *Wrong Navigation.(SR.FTA_2)*, *Wrong Navigation.(LSR.FTA_1)*, respectively. The countermeasure relationship in FPG shows what kinds of countermeasures we have for a system and can supplement to other similar faults. For instance, A safety countermeasure or safety guard *Secandary Sensor AVailable.(OR.ETA_1)*, which belongs to OR, can be supplemented to *Camera Sensor Failure.(SR.FMEA_0)* fault, which belongs to SR. Similarly, *Retrain Model.(SR.FMEA_0)* is supplemented to *Prediction model underperformance.(SR.FTA_0)* fault, and *Check Secondary Power source.(OR.ETA_1)* can be applied to *Electrical Power Failure.(LSR.FTA_1)* fault.

We also conducted a safety analysis for individual participating systems (SR, OR, LSR, and CS) and reported the identified faults as shown in Table 2. We identified 73 faults and 30 interaction-related faults for individual systems. However, we found 86 faults and 49 interaction-related faults by applying our composite safety analysis approach to HRRS. It enabled us to identify 18 percent more general faults and 63 percent more interaction faults than general safety analysis for HRRS. Last two columns (rightmost) of Table 2 show improvement in the finding of general faults and interaction-related faults, respectively. After all, SafeSoCPS considers more collaboration scenarios, and also it considers fault traceability within the system components and within the collaborating CPSs that led to identifying more faults. However, we did not consider much about the collaboration scenarios in general safety analysis for participating systems because these systems are deemed independent systems. However, we considered issues such as interoperability and heterogeneity and found some interaction-related faults.

A comparison of these two approaches (safety analysis for HRRS with SoCPSTracer and without SoCPSTracer) is also shown in Figure 8, where we see that for each system in HRRS (SR, OR, LSR, and CS), composite safety analysis found more faults than general safety analysis for HRRS. From Table 1 and Figure 8, it is clear that composite safety analysis, i.e., SafeSoCPS approach outperforms general safety analysis approach. Therefore, in response to **RQ1**, we can claim that our proposed approach, i.e., SafeSoCPS, finds more faults in the case of general faults and especially interaction-related faults.

Unlike individual systems, the SoCPS is exponentially more complex. The complexity arises due to their complex interactions and dependencies between participating systems. Our SoCPSTracer tool reduces this complexity in analyzing hazards for a network of SoCPS. The FPGs in SoCPSTracer are useful in investigating the impact of faults on the network of SoCPS. Safety engineers can easily find the propagation route of a fault with a single click on any node of a complex FPG.

## 6. Limitations of the Research

The SoCPSTracer tool that supports the SafeSoCPS approach presented herein is restricted to only FTA, FMEA, and ETA because hazard analysis artifacts produced from these three hazard analysis techniques can only be the input for the relationship detection algorithm. Therefore, it is worth mentioning that the SoCPSTracer is not applicable for hazard analysis artifacts obtained using other hazard analysis techniques. However, extension to support other hazard analysis techniques is straightforward. Meaning that safety engineers can consider contents of other hazard analysis techniques to apply same concept of relationships and extend it for their purpose. In particular, the relationship detector algorithm can be extended for the content of other hazard analysis techniques. After that it will be possible to easily create FPG by defining the relationships between the components of other hazard analysis techniques. Another limitation of this research is the threshold value (for purposes of the automatic identification of fault relationships) in Algorithm 1, which is subjective and can be determined by safety engineers according to their own application contexts.

SoCPSTracer can be applied in any domain aiming to trace the faults for collaborative CPSs. However, we have only applied SoCPSTracer to HRRS to prove its applicability. We did not apply it for other diverse domains. Therefore, the criticality of faults can differ from one application domain to another application domain. As mentioned, this tool aims to analyze safety for collaborative CPSs. However, it can also be used to analyze safety for a single CPS as well.

## 7. Conclusions

This paper introduces a composite safety analysis technique that presents a tool called SoCPSTracer to analyze safety for a SoCPS. SoCPSTracer consists of a safety analysis manager, traceability manager, and traceability presenter. The SoCPSTracer takes hazard artifacts as input and produces an FPG that shows the propagation of faults within a system and between systems of a SoCPS. Thus, the SoCPSTracer can be beneficial for safety engineers to trace the faults in the network of a SoCPS efficiently and economically. We used HRRS (a collaborative system) as a case study to validate our proposed approach. The results show that our proposed approach also called SafeSoCPS identifies more faults than general safety analysis techniques, i.e., SafeSoCPS approach identified 18 percent more general faults and 63 percent more interaction-related faults for SoCPSs. This is due to fact that SafeSoCPS is exclusively designed for collaborative systems and general safety analysis techniques do not consider the collaborative scenarios of CPSs.

Indications for future work include an extension of SoCPSTracer so that the faults can be traced back to the architectural documents and requirements documents to revise the architecture and requirements of the system. After receiving traceability information, efforts would be made to ensure each identified fault is mitigated to an acceptable level.

## Figures and Tables

**Figure 1 sensors-22-04474-f001:**
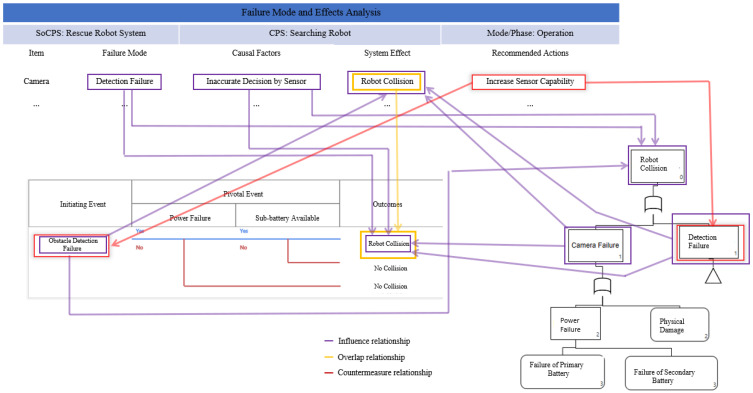
Example of relationships among hazard analysis artifacts (FMEA, FTA and ETA).

**Figure 2 sensors-22-04474-f002:**
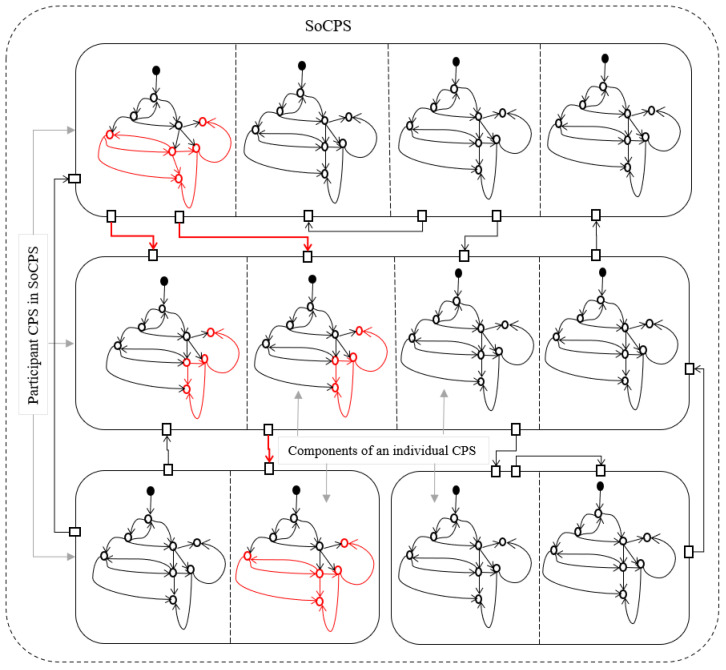
Fault propagation and traceability in SoCPS.

**Figure 3 sensors-22-04474-f003:**
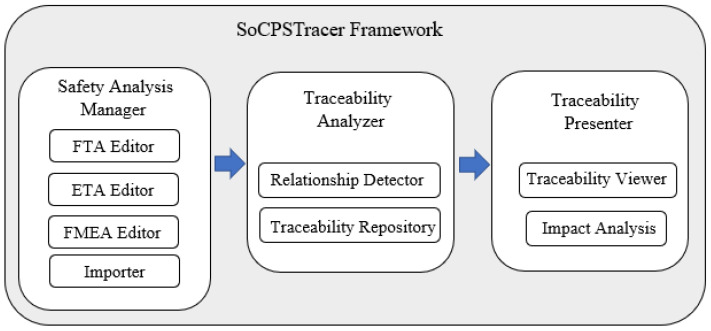
SoCPSTracer Framework.

**Figure 4 sensors-22-04474-f004:**
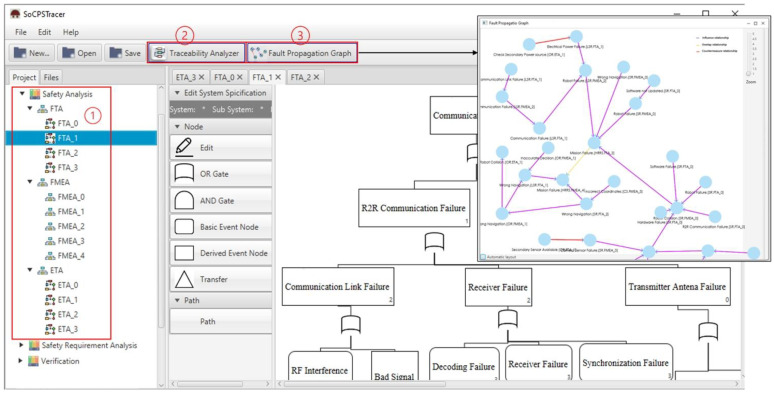
SoCPSTracer Tool View.

**Figure 5 sensors-22-04474-f005:**
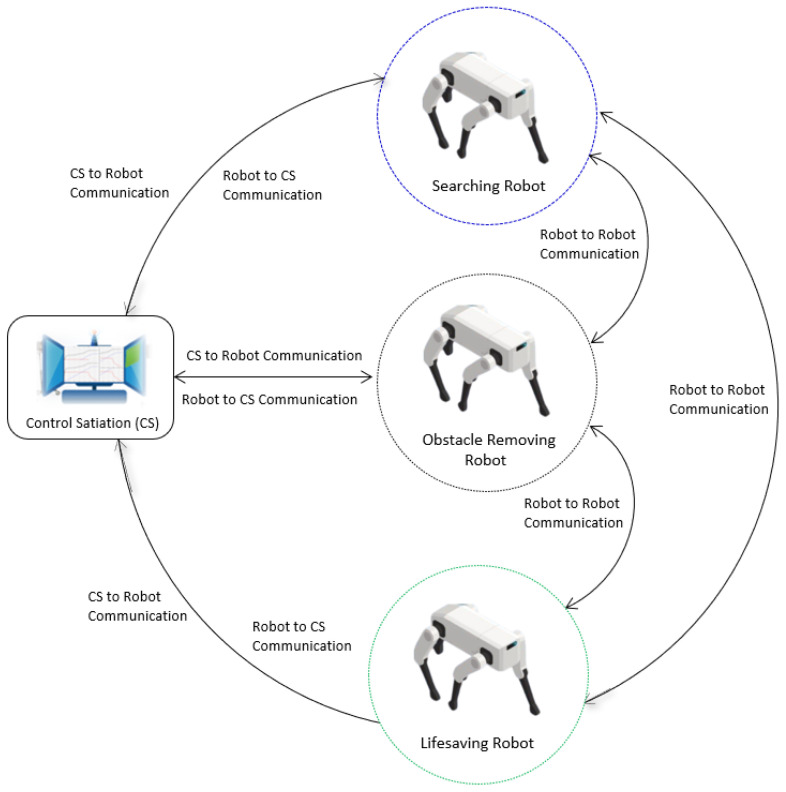
Human Rescue Robot System.

**Figure 6 sensors-22-04474-f006:**
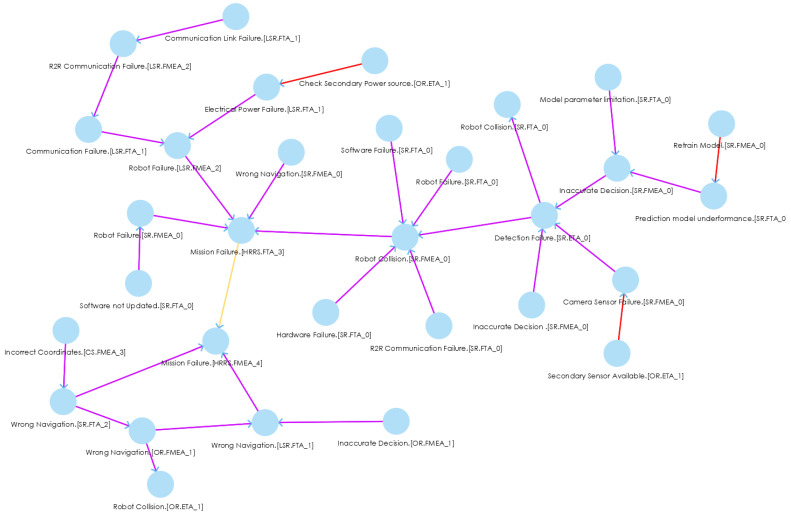
An excerpt of FPG for HRRS.

**Figure 7 sensors-22-04474-f007:**
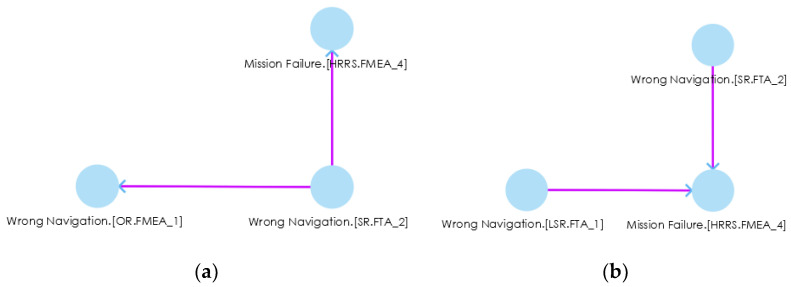
(**a**) Fault propagation route for *Wrong Navigation.(SR.FTA_2)*, (**b**) back traceability of a particular fault (*Mission Failure.(HRRS,FMEA_4*) in impact analysis.

**Figure 8 sensors-22-04474-f008:**
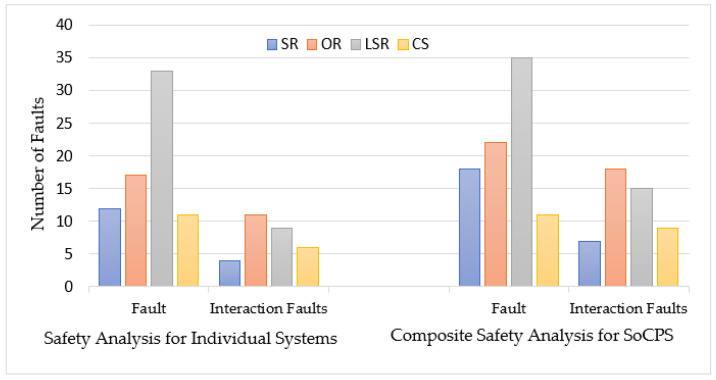
Comparision between traditional safety analysis and composite safety analysis for HRRS.

**Table 1 sensors-22-04474-t001:** Summary of related work.

Ref.	Composite Safety Analysis for SoS	Fault Traceability among/between SoS	Tool Support
[17]	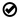	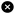	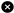
[19]	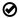	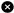	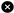
[20]	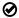	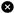	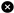
[21]	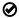	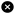	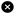
SafeSoCPS	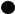	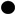	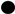

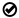
: Partially addressed; 
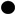
: fully addressed; 
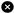
: not addressed.

**Table 2 sensors-22-04474-t002:** Safety analysis with and without composite safety analysis.

System	General Safety Analysis		Composite Safety Analysis			
	Faults (General)	Interaction-Related Faults	Faults (General)	Interaction-Related Faults	Improvement (Interaction-Related Faults)	Improvement (General Faults)
SR	12	4	18	7	50%	75%
OR	17	11	22	18	29%	64%
LSR	33	9	35	15	6%	67%
CS	11	6	11	9	0%	50%
**Total**	**73**	**30**	**86**	**49**	**18%**	**63%**

## Data Availability

Not applicable.

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
