# Peer review of "SafeSoCPS: A Composite Safety Analysis Approach for System of Cyber-Physical Systems"

_sensors, 2022, doi:10.3390/s22124474_

Round 1
Reviewer 1 Report
This manuscript introduces a composite safety analysis approach to analyze hazards for a network of SoCPS and trace the faults among participating systems through a fault propagation graph. My comments regarding the review of this manuscript are as follows:
1. The text in figures 3 and 7 is too blurred.
2. The formatting of some references needs to be adjusted, i.e., [13] and [14].
3. How to determine the threshold in Algorithm 1?
4. Each participant system needs to be analyzed before applying the SoCPSTracer. This operation is manual and complex for a network of SoCPS, and should be improved.
Author Response
Response to Reviewer 1 Comments
Manuscript ID: sensors-1738248
Title: SafeSoCPS: A Composite Safety Analysis Approach for System of Cyber-Physical Systems
First of all, we would like to thank you for precious time that you have taken to review our paper. We have addressed the points you have raised. Below is the response to your comments.
Point 1. This manuscript introduces a composite safety analysis approach to analyze hazards for a network of SoCPS and trace the faults among participating systems through a fault propagation graph. My comments regarding the review of this manuscript are as follows:
Point 1.1: The text in figures 3 and 7 is too blurred.
Response 1.1 [Modified]: We improved the figure 3 and 7.
Point 1.2: The formatting of some references needs to be adjusted, i.e., [13] and [14]..
Response 1.2 [Modified]: We have readjusted reference [13] and [14].
Point 1.3: How to determine the threshold in Algorithm 1?.
Response 1.3 [Modified]: We have made changes in the manuscript to explain this point. We made changes in section 6 (555-558) and in section 4 (from line 395 to 401). A safety engineer can assign a low threshold value if he wants to track down to minor faults, and conversely a high threshold if he wants to track only major faults.
Point 1.4: Each participant system needs to be analyzed before applying the SoCPSTracer. This operation is manual and complex for a network of SoCPS and should be improved.
Response 1.4:[Replied] The SoCPSTracer can also be used to analyse system individually. However, the main application of SoCPSTracer is to analyse safety for SoCPSs. The final goal is the fault traceability in the SoCPSs. We also mentioned it on the last paragraph of section 6 (line 562-564)

Reviewer 2 Report
This paper introduces a composite safety analysis technique and a tool, SoCPSTracer, to analyse safety for a SoCPS (System of Cyber Physical Systems).
The subject is relevant, the comparison to related work is correct, the research well presented and it includes up to date material and a well presented case study from which to learn (and teach in class) the safety analysis.
The paper is interesting for the professionals working on safety. Having said that, I recommend the following revisions are necessary
1. Details of the availability, computer system requirements and application possibilities of SoCPSTracer should be provided.
2. Section 6, Threats to Validity, should be termed "limitations of the Research" and expanded to clearly identify the limitations. Presently it is more of a general description like "However, extension to support other hazard analysis techniques is straightforward" . This should be explained in clear detail to be at the same level as others section of the paper (e.g. section 5.2)
3. The presentation after line 492 should be separated into a new "discussion of results" section.
4. The Conclusion section should better highlight the differences and advantages of the proposed approach over existing ones (based on the "discussion of results" suggested above)
4. Editing should revised for readability:
Example 1: lines 520 to 524, the reader needs much effort to understand the answer to RQ2.
Example 2: Line 222-224 is difficult to understand . Are the authors referencing some stage of the research presented in this paper or some previous publication that some of the authors participated in? (the latter seems the correct answer)
There are other. As stated, general editing should be revised.
Author Response
Response to Reviewer 2 Comments
Manuscript ID: sensors-1738248
Title: SafeSoCPS: A Composite Safety Analysis Approach for System of Cyber-Physical Systems
First of all, we would like to thank you for precious time that you have taken to review our paper. We have addressed the points you have raised. Below is the response to your comments.
Point 1. The paper is interesting for the professionals working on safety. Having said that, I recommend the following revisions are necessary:
Point 1.1. Details of the availability, computer system requirements and application possibilities of SoCPSTracer should be provided.
Response 1.1 [Modified]: We have made changes to manuscript (Section 4) to address this point.
“ SoCPSTracer is implemented in Java and JavaFx was used to develop user interface. The system was equipped with core i7 processors and 32 GB RAM for the experimental setup. Additionally, NVIDIA GeForce RTX2060 GPU was added to the system to faster the computational process, and better visualization of data on FPG. Data visualization on FPG is supported by an opensource java library called smartgraph[1] that supports directed and undirected graph generation. We customized the smartgraph library to visualize the data on FPG according to our requirements.”
Regarding the availability of the system, we are making documentation and it will be available on GitHub very soon.
Point 1.2. Section 6, Threats to Validity, should be termed "limitations of the Research" and expanded to clearly identify the limitations. Presently it is more of a general description like "However, extension to support other hazard analysis techniques is straightforward" . This should be explained in clear detail to be at the same level as others section of the paper (e.g., section 5.2).
Response 1.2 [Modified]: We have made changes to the manuscript. The changes can be seen in section 6.
Point 1. 3. The presentation after line 492 should be separated into a new "discussion of results" section.
Response 1.3. [Modified]: We have separated it from line 492 and named it “discussion of results”.
Point 1.4. The Conclusion section should better highlight the differences and advantages of the proposed approach over existing ones (based on the "discussion of results" suggested above).
Response 1.4. [Modified]: We have modified the conclusion section and adjusted the reviewer’s comments. Please refer to the section 7 (conclusion section).
Point 1.5. Editing should revised for readability:
Example 1: lines 520 to 524, the reader needs much effort to understand the answer to RQ1.
Response 1.5a. [Modified]: A comparison of these two approaches (safety analysis for HRRS with SoCPSTracer and without SoCPSTracer) is also shown in Figure 8, where we see that for each system in HRRS (SR, OR, LSR, and CS), composite safety analysis found more faults than general safety analysis for HRRS. From Table 1 and Figure 8, it is clear that composite safety analysis i.e., SafeSoCPS approach outperforms general safety analysis approach. Therefore, in response to RQ1, we can claim that our proposed approach, i.e., SafeSoCPS, finds more faults in the case of general faults and especially interaction-related faults.
Example 2: Line 222-224 is difficult to understand . Are the authors referencing some stage of the research presented in this paper or some previous publication that some of the authors participated in? (the latter seems the correct answer).
Response 1.5b. [Modified]: You are right. We published a paper about basic ideas of the content-based relationship in the reference [24]. However, we present an improved and precise version of those defined relationships in this article.
There are other. As stated, general editing should be revised.
Response 1.5c. [Replied]: We have made changes to the manuscript to address above comments. We did a proofread and corrected a number general editing to increase readability.
[1] https://github.com/brunomnsilva/JavaFXSmartGraph

Round 2
Reviewer 1 Report
I have no further comments.